# COVID-19 Vaccine Hesitancy among Population in Jazan Region of Saudi Arabia

**DOI:** 10.3390/healthcare11233051

**Published:** 2023-11-27

**Authors:** Manal Almalki, Mohammed Kotb Sultan, Mohammed Abbas, Ajiad Alhazmi, Yasser Hassan, Joe Varghese

**Affiliations:** 1Department of Health Informatics, College of Public Health and Tropical Medicine, Jazan University, Jazan 45142, Saudi Arabia; almalki@jazanu.edu.sa; 2Department of Community Medicine, Faculty of Medicine, Ain Shams University, Cairo 11566, Egypt; prof_mohammedkotb@yahoo.com; 3Department of Epidemiology, College of Public Health and Tropical Medicine, Jazan University, Jazan 45142, Saudi Arabia; moabbas032@gmail.com (M.A.); 201906225@stu.jazanu.edu.sa (A.A.); yahassan@jazanu.edu.sa (Y.H.)

**Keywords:** COVID-19, vaccine efficacy, rural population, vaccine acceptance, vaccine hesitancy, Saudi Arabia

## Abstract

COVID-19 vaccine acceptance and refusal vary across countries and among different socio-demographic groups. This study investigates hesitancy related to the COVID-19 vaccine and the associated factors in the rural-community-dominated Jazan Province, Saudi Arabia. A cross-sectional study through an online questionnaire was conducted from February to April 2021 to investigate the extent of vaccine hesitancy related to the COVID-19 vaccine and the associated factors in the Jazan region. A Chi-squared test and post hoc analysis were conducted to analyze the statistical significance of the association between variables. Of the 569 participants who completed the online questionnaire, the majority were males (81.5%) and had a university education (72.6%). Of the participants, more than one-third (36.9%) were hesitant to vaccinate. Concern about adverse side effects following vaccination was the most reported reason for vaccine hesitancy (42.6%), followed by beliefs that the vaccine was unsafe or ineffective (15.5%). The data analysis revealed that people who lived in cities in Jazan Province or those who did not have a family history of COVID-19 infection were more likely to be vaccine hesitant. It is more important than ever to develop and implement community-based strategies to address vaccine hesitancy, especially in rural areas.

## 1. Introduction

The COVID-19 pandemic has highlighted the importance of vaccine acceptance and uptake in controlling the spread of disease. In March 2020, the World Health Organization (WHO) declared COVID-19 a pandemic [1]. By May 2023, the WHO declared the end of the pandemic as a “global health emergency”, but it had already affected over 765 million people worldwide and caused nearly seven million deaths [2,3]. Governments around the world took urgent action to detect and contain the virus, including travel restrictions, quarantines, mass vaccination campaigns, mass screening programs, and free COVID-19 treatment for all citizens and residents [4]. COVID-19 vaccines were proven to be very effective in preventing serious illness, hospitalization, and death from COVID-19. However, vaccine acceptance and uptake have been variable around the world. In some countries, vaccination rates have been very high, whereas they have been much lower in others.

In Saudi Arabia, before COVID-19 vaccines became available to the public, the Ministry of Health (MoH) made several preparations to immunize the entire population [5]. Saudi Arabia recognized the following COVID-19 vaccines: Oxford-AstraZeneca, Pfizer/BioNTech, Moderna, Johnson & Johnson, Sinopharm, Sinovac, Covaxin, and Sputnik V. Vaccines were provided free of charge to all citizens and residents. The initial stage of the COVID-19 vaccination campaign prioritized the most vulnerable, followed by the general population.

To encourage vaccine uptake among the population, the Saudi Ministry of Health (MoH) launched the “Awlawia” service, which means priority; this service enabled people of older ages to receive the vaccine at any vaccination facility without a wait or a reservation [6]. Additionally, vaccination campaigns have used social media networks such as Twitter to urge communities to engage and comply with vaccination measures. For example, a campaign with the hashtag “Natawan ma Natahawan”, which means “we cooperate, not relax” was launched on the social media platform Twitter to spread information on COVID-19 vaccination initiatives [7]. Furthermore, Saudi MoH effectively utilized contact tracing technologies such as the Tawakkalna App to ensure high vaccination coverage among the population [8,9]. As of 2 May 2023, only 77.67% of people in Saudi Arabia have been vaccinated with at least one dose of the COVID-19 vaccine, despite the government’s efforts to promote vaccination [3].

Vaccine hesitancy is the delay in acceptance or refusal of vaccination despite the availability of vaccination services. Previous research has shown that vaccine hesitancy is more likely to increase during special vaccination initiatives, such as the COVID-19 vaccination campaign than during routine immunization campaigns [10,11,12]. Studies have shown that vaccine hesitancy can become a social phenomenon if it becomes widespread and can lead to the failure of vaccination programs, as many people in one’s social network may not recommend vaccination [13].

A systematic review published in 2023 and based on 24 articles estimated the average COVID-19 vaccine hesitancy rate to be 12% [14]. According to this systematic review, individuals with a high socioeconomic status, history of previous vaccination, and medical background had a lower rate of COVID-19 vaccination refusal. Another global survey on COVID-19 vaccine acceptance found that acceptance rates ranged from nearly 90% in China to less than 55% in Russia [15]. This study reported higher income and education and higher trust in government as positive predictors of COVID-19 vaccine acceptance. A large multi-country study found that vaccine hesitancy among Arab-speaking people is high (83.3%) [16]. Moreover, this study also reported a significant association between conspiracy beliefs and COVID-19 vaccine hesitancy. A study conducted in Saudi Arabia found that the prevalence of COVID-19 vaccine hesitancy is 20.6% [17]. Concerns about vaccine safety and efficacy, belief in conspiracy theories, and the role of social media were identified in this study as the leading factors for vaccine hesitancy.

Our study aims to investigate the extent of vaccine hesitancy related to the COVID-19 vaccine and the associated factors in the Jazan Region of the Kingdom of Saudi Arabia (KSA). Although Jazan Province is the smallest in Saudi Arabia, it has the highest population density. It is divided into 13 governorates and 31 centers with at least 2500 inhabitants [18]. It also has around 80 islands on the Red Sea, shares 330 km of coastline on the west, and shares borders with the Republic of Yemen on the south and southeast. Jazan Province has a higher proportion of rural population than other regions of Saudi Arabia [19]. Many studies have revealed a disparity between rural and urban areas in the prevalence of vaccine reluctance [20,21]. According to these studies, rural societies often have less access to accurate information about vaccines due to limited internet access, less access to healthcare providers, and cultural barriers.

This study is expected to provide policymakers and MoH with insights into the root causes of vaccine hesitancy and rejection and the factors that contribute to it. This information is essential for improving vaccine coverage in Saudi Arabia. As new variants may still develop and cause new outbreaks, we need a strategy to tackle the factors deterring the uptake of COVID-19 vaccine booster doses to actively engage the Saudi population and raise awareness of the importance of receiving the booster shot.

## 2. Materials and Methods

This study was carried out in Jazan Province. It is located in the southwest corner of the KSA between longitude 42.7076° E and latitude 17.4751° N, with a total land area of approximately 40,457 km^2^ and a total population of approximately 1.8 million. The Kingdom of Saudi Arabia reported the first case of COVID-19 infection in March 2020. When we began collecting data for this study, there were over 373,368 confirmed COVID-19 cases and 6441 deaths in Saudi Arabia [22].

The minimum calculated sample size for the study, assuming a prevalence of 50%, a confidence interval of 95%, and an error of 5%, was 385.

An online questionnaire was developed by using Google Forms. The draft questionnaire was initially developed by the researchers and validated with the help of two independent public health experts. The draft tool was translated from English to Arabic and then back to English by an independent translator to ensure the accuracy of the translation. A pilot study was conducted among eight volunteers to ensure clarity of the questions to participants, and survey questions were modified based on feedback.

Social media platforms such as Twitter, WhatsApp, and Telegram were used to recruit participants from 17 February to 15 April 2021, and this period matched the second wave of COVID-19 in Saudi Arabia. COVID-19 vaccination had already begun, with over half a million doses delivered by the time the online survey started [22].

The opening page of the Google form carried an informed consent, which included an explanation of the study objectives and processes and guaranteed anonymity to participants during the study processes. The study was approved by the research committee of the Jazan University, Saudi Arabia (REC 41/5/144).

The questionnaire comprised 21 questions, which were divided into 6 demographic questions and 15 questions related to the aim of the study, which included participants’ willingness or unwillingness to receive vaccination, their source of information about COVID-19 vaccination, and the reasons for vaccination hesitation. At the beginning of the survey, residence information was collected to ensure that only participants from the Jazan region joined the study. The names of the participants were not recorded to ensure the anonymity of the participants.

The data collected were imported from Google Forms into the SPSS program (IBM Statistical Package for Social Studies) version 20.0. Frequency tables and Chi-squared tests were conducted to analyze the data obtained. We used post hoc analysis by employing the Bonferroni method, as the groups compared were more than two to identify the significantly different group. The Bonferroni method adjusted the *p* value and is calculated by dividing the conventional *p* value (0.05) by the number of comparisons to control the family-wise type I error [23].

## 3. Results

Five hundred and sixty-nine participants completed the questionnaire. As shown in Table 1, there were more males than females, and the majority were Saudi citizens (96.7%). More than half of the participants were between 20 and 29 years old. Approximately half of the participants lived in the cities and had a university level of education.

The survey assessed the distribution of study participants according to their acceptance of COVID-19 vaccination. The data in Table 1 shows that 359 participants (63.1%) agreed to receive the vaccine, whereas 210 participants (36.9%) were hesitant to COVID-19 vaccination.

Table 1 shows that vaccine acceptance and factors, including gender, age, and marital status, could not be linked statistically. However, residents of towns and islands were statistically significantly more likely to accept vaccinations than those living in cities. It also shows that the three educational level groups had a statistically significant difference in their attitude towards the COVID-19 vaccine.

Table 2 shows that although the omnibus Chi-squared analysis revealed a significant difference between the three educational groups (*p* value = 0.037), no group is significantly different when its *p* value is compared to the Bonferroni-corrected *p* value (0.0083). It is due to the conservative nature of the Bonferroni correction, which controls family-wise type I errors. However, the cost of using the Bonferroni correction is a reduction in statistical power, which is the ability to detect a true effect when it exists. Therefore, we used multiple two-by-two analyses as an alternative way to identify the statistically different group. Multiple two-by-two analyses showed that the middle school group is statistically more hesitant to accept the COVID-19 vaccine than the high school and university groups (*p* values are 0.012 and 0.047, respectively) (Appendix A). In contrast, the high school and the university groups are statistically indifferent (*p* value = 0.161).

This survey also captured the perception of study participants on various aspects of the COVID-19 vaccination program (Table 3 and Table 4). Regarding the participants’ primary sources of vaccine information, the data show that participants who relied on social media as the primary source of information are more likely to show vaccine hesitancy than those who relied on mass media for vaccine information (Table 4). Likewise, the study participants who agree to receive vaccines have a positive perception of the vaccine’s ability to reduce COVID-19 infection. However, people who were hesitant to take the vaccines had less faith in the ability of vaccines to reduce the incidence of infection. Table 4 also shows that participants who do not believe that physical protective measures alone could lower the chance of infection were more likely to accept vaccination than those who believed it. Those who were not concerned about the side-effects of the COVID-19 vaccine were also more willing to accept vaccination. Additionally, participants whose family members had previously been infected with COVID-19 were more likely to accept vaccination. These findings are statistically significant.

Out of the 210 study participants who showed hesitancy to receive the COVID-19 vaccine, 155 responded to the questions on the main reason for their vaccine hesitancy. Table 5 shows that the majority of the respondents expressed fear of vaccine side effects (42.6%), followed by concerns about the safety of the vaccine (15.5%), and concerns about the rapid development of the vaccine (13.5%).

## 4. Discussion

In our study, an online questionnaire was used to determine the public acceptability of the COVID-19 vaccination and the causes of vaccine reluctance in the Jazan region of Saudi Arabia. A total of 63.1% of participants indicated that they decided to take the vaccine. However, the survey reveals that 36.9% of the individuals are unsure about getting COVID-19 vaccine. Our study showed that vaccine acceptance and factors, including gender, age, and marital status, could not be linked statistically. However, residents of towns and islands were statistically significantly more likely to accept vaccinations than those living in cities. Similarly, those with high school- and university-level education accepted the vaccines statistically more than those with only middle school-level education.

The overall COVID-19 vaccine acceptance and hesitancy shown in this study can be compared with the findings of similar studies conducted in Saudi Arabia and elsewhere. Although several studies from Saudi Arabia have reported on vaccine acceptance and hesitancy, the reported vaccine acceptance rates vary significantly. One nationwide study found a lower COVID-19 vaccine acceptance rate of 27.3%, whereas another found a much higher vaccine acceptance rate of 79.24% [24,25]. However, most studies report vaccine acceptance rates similar to the rate observed in this study [26,27,28,29,30]. The variation in vaccine acceptance rates between studies may be due to several factors, including the study design, the sample population, and the period in which the study was conducted. It is important to remember that vaccine acceptance can change over time.

This study, when compared to other similar studies in the GCC region, found that a lower level of vaccine acceptance was seen in other countries, such as Kuwait (23.6%) and Jordan (28.4%) [31]. Our results are also consistent with several other international estimates. For example, a study conducted in Ireland and the United Kingdom found that 26% of people in Ireland and 25% in the United Kingdom were reluctant to take the vaccine. Only 65% of the Irish population and 69% of the UK population were fully willing to accept a COVID-19 vaccine [32]. A recent global survey conducted in June 2020 involving 19 countries reported that acceptance rates ranged from almost 90% in China to less than 55% in Russia [15].

Not many studies on COVID-19 vaccine acceptance in Saudi Arabia focused on rural populations. Only one previous study has been conducted among the general population of Jazan Province. The study, published in 2021 by Alamer et al. (2021), found that 67% of participants had positive perceptions toward COVID-19 vaccines [33]. Two other studies conducted among Jazan University students found 83.6% and 90.4% as the COVID-19 vaccine acceptance rate [34,35]. The findings of these studies suggest that there is a high level of acceptance of COVID-19 vaccines in Jazan Province. However, it is essential to note that these studies were conducted at different times and with different populations. Therefore, it is not possible to compare the findings directly with our study.

Like in this study, the studies from other parts of the world also have noted high acceptance of COVID-19 vaccination in rural regions [36,37]. One possible reason for lower hesitancy to vaccination in rural areas is the reduced access to social media that spread messages against vaccination in rural areas [38].

One of the findings of this study is lower acceptance of the COVID-19 vaccine among those having middle school education when compared with respondents having high school- and university-level education. Other studies on COVID-19 vaccine acceptance in Saudi Arabia also conform to this result, which reported lower vaccine acceptance among less educated [26,28]. However, a study by Magadmi and Kamel found lower acceptance of COVID-19 vaccination among the university educated in Saudi Arabia [36]. Further studies are required to understand the role of education in vaccine acceptance.

Hesitancy to vaccination is as old as vaccination programs, and it represents a serious threat to global health [39,40]. In light of a pandemic that has caused significant health and socio-economic devastation, a remarkable positive response to vaccination initiatives is expected. However, experts have warned that the pandemic, which generates fear, anxiety, and uncertainty, could also create a suitable environment for the emergence of conspiracy theories [41,42,43]. Our study, which analyzed the pattern of hesitancy, sheds more light on this. A deep look into the vaccination-related perceptions of study participants shows that the fear of vaccine side effects, rather than the vaccine’s ineffectiveness, is a dominant factor in the decisions of study participants who have reported hesitancy towards vaccinations. Several other studies on the acceptance of the COVID-19 vaccine also share the same findings [44,45].

Community engagement can be a significant factor in the acceptability of vaccines. Specific actions are required to tackle vaccine hesitancy in predominantly rural societies like Jazan. A recent study on the perspectives of Jazan University students toward adherence to COVID-19 preventive measures identified misconceptions as one of the barriers [46]. Vaccine messaging should be tailored to the specific cultural context of the community, which means using language and images that are familiar to the community and that address the specific concerns that people in the community have about vaccines. In addition, the community’s acceptance of vaccines would increase if more information about vaccine safety and efficacy were available. To combat vaccine hesitancy, the public should be “immunized” against misinformation about the vaccine’s side-effects, preferably by a trusted, centralized source of information such as mass media including radio and television. The role of vaccinators and frontline health workers in addressing the concerns of people should be acknowledged because they are essential to the success of vaccination programs. They are the ones who interact with people on a daily basis and build trust with them [12,47].

Our study has several limitations. Firstly, it is a cross-sectional study, which depicts a picture of the community response towards COVID-19 vaccination at the time of the study. Vaccine preferences are dynamic and have a solid connection to people’s perception of the risk of disease and the risk of vaccine side effects [48]. These factors can change as the pandemic and vaccination programs advance in the future. Secondly, the study used a non-probability sampling method for the selection of participants, which would have affected the external validity of the study. There is also a possibility of social desirability bias that might have motivated the study participants to answer questions in a socially acceptable or desirable way. However, we do not expect a significant influence of this bias in our study as the data were collected through an anonymous self-reported questionnaire.

## 5. Conclusions

Although vaccination is one of the best strategies to combat the COVID-19 pandemic, vaccine hesitancy can significantly undermine future plans. This study, in which approximately one-third of the study participants expressed vaccine hesitancy or rejected COVID-19 vaccines, should alert health authorities to the remedial actions needed to ensure high levels of vaccine coverage.

Building confidence in COVID-19 vaccines is essential to control the pandemic. Several factors, particularly in rural areas, such as community engagement and the vaccine’s perceived effectiveness and safety can contribute to increasing trust in vaccinations. Authorities should treat people’s worries about vaccines with respect and understanding. Efforts should be made to listen to their concerns, give them accurate information, and help them to feel safe and comfortable. Widespread public awareness campaigns on the benefits of COVID-19 vaccination, which also assure the safety of vaccinations, are required. Successful use of the mass media or promotions using prominent personalities can go a long way in reassuring apprehensions against COVID-19 vaccines in the community.

Social media is a powerful tool that can be used to spread information about vaccines, both positive and negative. In Saudi Arabia, social media is widely used, and it is likely to play a role in shaping public opinion about vaccines. Passive surveillance on social media to check for misinformation or falsified information on vaccination programs can help in identifying them earlier and in developing counter strategies.

Further research is needed to learn more about vaccine hesitancy in rural areas like Jazan Province, Saudi Arabia, as the vaccination program continues and the pandemic scenario changes. Studies with larger and more representative samples can provide more accurate insights into the prevalence of vaccine hesitancy in Saudi Arabia, the factors that contribute to vaccine hesitancy, and the most effective ways to address vaccine hesitancy. Likewise, more research is needed to understand how social media is being used to spread information about vaccines in Saudi Arabia and how this information is influencing vaccine acceptability and hesitancy.

## Figures and Tables

**Table 1 healthcare-11-03051-t001:** Prevalence of COVID-19 vaccine hesitancy according to demographic characteristics of the participants.

Variables		Hesitant to Receive COVID-19 Vaccine		Willing to Receive COVID-19 Vaccine		Total	*p*-Value *
		*n*	%	*n*	%		
Gender	Male	173	37.3	291	62.7	464	0.695
	Female	37	35.2	68	64.8	105	
Age (years)	<20	31	45.6	37	54.4	68	0.389
	20–29	106	34.6	200	65.4	306	
	30–39	34	33.7	67	66.3	101	
	40–49	27	45.0	33	55.0	60	
	50–59	10	37.0	17	63.0	27	
	≥60	2	28.6	5	71.4	7	
Educational level	Middle School	11	61.1	7	38.9	18	0.037
	High School	43	31.2	95	68.8	138	
	University level education	156	37.8	257	62.2	413	
Marital status	Single	137	38.1	223	61.9	360	0.456
	Married	73	34.9	136	65.1	209	
Residency	Cities	127	41.4	180	58.6	307	0.017
	Towns and islands	83	31.7	179	68.3	262	
Total		210	36.9	359	63.1	569	

* Chi-square test.

**Table 2 healthcare-11-03051-t002:** Post hoc analysis for the significant Chi-squared results of the educational groups by using Bonferroni method.

Variables			Hesitant to Receive COVID-19 Vaccine	Willing to Receive COVID-19 Vaccine	Total	*p*-Value
Educational level	Middle School	*n* (%)	11 (61.1)	7 (38.9)	18 (100)	0.037
		Adjusted Residual	2.2	−2.2		
		*p*-value *	0.031	0.031		
	High School	*n* (%)	43 (31.2)	95 (68.8)	138 (100)	
		Adjusted Residual	−1.6	1.6		
		*p*-value *	0.108	0.108		
	University level education	*n* (%)	156 (37.8)	257 (62.2)	413 (100)	
		Adjusted Residual	0.7	−0.7		
		*p*-value *	0.5	0.5		
	Total		210 (36.9%)	359 (63.1%)	569 (100%)	

* The Bonferroni-corrected *p* value = 0.05/6 = 0.0083.

**Table 3 healthcare-11-03051-t003:** Prevalence of COVID-19 vaccine hesitancy according to source of COVID-19 vaccine information, perceived risk of COVID-19 infection, perception of risk of COVID-19 vaccination, and health conditions of the study participants.

Variables		Hesitant to Receive COVID-19 Vaccine		Willing to Receive COVID-19 Vaccine		Total	*p*-Value *
		*n*	%	*n*	%		
Source of COVID-19 vaccine information	Mass media (radio/TV)	143	32.4	298	67.6	441	<0.001
	Social media	59	54.1	50	45.9	109	
	Family members	3	25.0	9	75.0	12	
	Newspapers and magazines	5	71.4	2	28.6	7	
Do you think that you are at risk of COVID-19 infection?	Yes	112	33.0	227	67.0	339	0.068
	No	47	42.7	63	57.3	110	
	Do not know	51	42.5	69	57.5	120	
Do you think that COVID-19 vaccine reduces the risk of infection?	Yes	95	23.3	312	76.7	407	<0.001
	No	29	82.9	6	17.1	35	
	Do not know	86	67.7	41	32.3	127	
Do you think you can protect yourself from COVID-19 infection solely through physical protective measures?	Yes	108	46.6	124	53.4	232	<0.001
	No	32	19.2	135	80.8	167	
	Do not know	70	41.2	100	58.8	170	
Are you concerned with the risk of COVID-19 vaccine adverse effects	Yes	166	49.4	170	50.6	336	<0.001
	No	14	11.7	106	88.3	120	
	Do not know	30	26.5	83	73.5	113	
Presence of chronic diseases	Yes	38	42.7	51	57.3	89	0.218
	No	172	35.8	308	64.2	480	
Previous personal infection with COVID-19	Yes	35	40.7	51	59.3	86	0.429
	No	175	36.2	308	63.8	483	
Family members’ previous infection with COVID-19 virus	Yes	116	33.5	230	66.5	346	0.037
	No	94	42.2	129	57.8	223	
Total		210	36.9	359	63.1	569	

* Chi-squared test.

**Table 4 healthcare-11-03051-t004:** Post hoc analysis for the significant Chi-squared results in Table 3 by using Bonferroni method.

Source of COVID-19 vaccine information	Mass media	*n* (%)	143 (32.4%)	298 (67.6%)	441 (100%)
		Adjusted Residual	−4.1	4.1	
		*p* value *	<0.001	<0.001	
	Social media	*n* (%)	59 (54.1%)	50 (45.9%)	109 (100%)
		Adjusted Residual	4.1	−4.1	
		*p* value *	<0.001	<0.001	
	Family members	*n* (%)	3 (25.0%)	9 (75.0%)	12 (100%)
		Adjusted Residual	−0.9	0.9	
		*p* value *	0.388	0.388	
	Newspapers and magazines	*n* (%)	5 (71.4%)	2 (28.6%)	7 (100%)
		Adjusted Residual	1.9	−1.9	
		*p* value *	0.057	0.057	
	Total	*n* (%)	210 (36.9%)	359 (63.1%)	569 (100%)
Do you think that COVID-19 vaccine reduces the risk of infection?	Yes	*n* (%)	95 (23.3%)	312 (76.7%)	407 (100%)
		Adjusted Residual	−10.6	10.6	
		*p* Value **	<0.001	<0.001	
	No	*n* (%)	29 (82.9%)	6 (17.1%)	35 (100%)
		Adjusted Residual	5.8	−5.8	
		*p* Value **	<0.001	<0.001	
	Do not Know	*n* (%)	86 (67.7%)	41 (32.3%)	127 (100%)
		Adjusted Residual	8.2	−8.2	
		*p* Value **	<0.001	<0.001	
	Total	*n* (%)	210 (36.9%)	359 (63.1%)	569 (100%)
Do you think you can protect yourself from COVID-19 infection solely through physical protective measures?	Yes	*n* (%)	108 (46.6%)	124 (53.4%)	232 (100%)
		Adjusted Residual	4.0	−4.0	
		*p* Value **	<0.001	<0.001	
	No	*n* (%)	32 (19.2%)	135 (80.8%)	167 (100%)
		Adjusted Residual	−5.7	5.7	
		*p* Value **	<0.001	<0.001	
	Do not Know	*n* (%)	70 (41.2%)	100 (58.8%)	170 (100%)
		Adjusted Residual	1.4	−1.4	
		*p* Value **	0.168	0.168	
	Total	*n* (%)	210 (36.9%)	359 (63.1%)	569 (100%)
Are you concerned with the risk of COVID-19 vaccine adverse effects	Yes	*n* (%)	166 (49.4%)	170 (50.6%)	336 (100%)
		Adjusted Residual	7.4	−7.4	
		Chi-square	55.0	55.0	
		*p* Value **	<0.001	<0.001	
	No	*n* (%)	14 (11.7%)	106 (88.3%)	120 (100%)
		Adjusted Residual	−6.5	6.5	
		*p* Value **	<0.001	<0.001	
	Do not Know	*n* (%)	30 (26.5%)	83 (73.5%)	113 (100%)
		Adjusted Residual	−2.5	2.5	
		*p* Value **	0.011	0.011	
	Total	*n* (%)	210 (36.9%)	359 (63.1%)	569 (100%)

* The Bonferroni-corrected *p* value = 0.05/8 = 0.0063, ** The Bonferroni-corrected *p* value = 0.05/6 = 0.0083.

**Table 5 healthcare-11-03051-t005:** Reported main reason for COVID-19 vaccine hesitation.

Reasons	*n*	%
Afraid of vaccine side-effects	66	42.6
The vaccine not safe	24	15.5
Vaccine was developed in short period	21	13.5
Suffer from chronic diseases	14	9.0
Not enough information about vaccine safety	7	4.5
Belief in vaccine conspiracy	6	3.9
Scared of the vaccine injection	6	3.9
Waiting for more evidence of the vaccine safety	5	3.2
Prefer to follow behavioral protective measures	3	1.9
Other reasons including pregnancy	3	1.9
Total	155 *	100

* Fifty-five participants who were hesitant did not specify the reasons of hesitancy.

## Data Availability

Data are contained within the article and Appendix A.

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
