# Peer review of "COVID-19 Vaccine Hesitancy among Population in Jazan Region of Saudi Arabia"

_healthcare, 2023, doi:10.3390/healthcare11233051_

Round 1

Reviewer 1 Report

Comments and Suggestions for Authors

The authors of the manuscript titled “COVID-19 Vaccine Hesitancy among Population in Rural Regions in Saudi Arabia” performed an online questionnaire during the period from February to April 2021 to investigate the extent of vaccine hesitancy related COVID-19 vaccine in Jazan region. According to the authors, “people who lived in cities in the Jazan Provence, and who had a family history of COVID-19 infection were significantly more likely to be reluctant to receive the vaccine.” Among the manuscript's other conclusions.

I take this opportunity to thank the authors for their efforts in conducting this study. However, there are some minor issues and a few major issues in the manuscript that must be addressed.

Minor issues:

1.      In lines 67-68 “out of 100 people, only 77.67 have been vaccinated with at least one dose per 100 people in the Kingdom” some words repeated.

2.      Line 78 “About 86.9% of parents around the world were hesitant in vaccinating their children.” Reference missing.

3.      Lines 81-84 reference is missing.

4.      Lines 94 - 104 better move to the conclusion part.

5.      The procedure for obtaining informed consent from the online participants must be explained in the materials and method section.

6.      The IRB/ethical approval letter number must be listed in the materials and method section.

7.      How was the participants' confidentiality maintained after submitting the response to the questionnaire?

8.      In line 127, the authors mentioned that “the majority were Saudi citizens.” While there is no indication in any of the tables what were the numbers or percentages of the nationalities of the participants?

9.      More recent references must be used in the discussion part.

10.  The authors claim in lines 210-211 that they “could not find any previous study of COVID-19 vaccine acceptance in Saudi Arabia which had a focus on rural populations.” However, in the previous paragraph, they cite at least three references on COVID-19 vaccine hesitancy in Jazan (the same studied province in this manuscript). These references are 34 & 35 and 36!!

11.  Reference number 48 in line 239 is not listed in the references section.

12.  Text in lines 242-245 is probably taken from a previous reviewer’s comments. Better to omit it. “Authors should discuss the results and how they can be interpreted from the perspective of previous studies and of the working hypotheses. The findings and their implications should be discussed in the broadest context possible. Future research directions may also be highlighted.”

Major points:

1.      The authors’ interpretation of the significant values resulted from using the Chi-square analysis used in tables one and two when the compared groups were 3X2 (such as Educational level and vaccine hesitancy in table 1) and 4X2 (e.g. source of COVID-19 vaccine information and COVID-19 vaccine hesitancy in table 2) the authors concluded that “those with a high school education were more willing to accept vaccines than those with a university degree or educated only up to middle school. The difference was statistically significant”. Probably because the percentage of the “willing to receive COVID-19 vaccine” group was bigger than the University level. However, although the” overall” difference between the three groups was significant, a post hoc test must be performed to identify which group is significantly different from the other. The same applies to the data in Table 2. Therefore, I suggest that the statistical analysis and data interpretation must be significantly revised.

2.      The authors claim in the abstract that “people who lived in cities in the Jazan Provence, and who had a family history of COVID-19 infection were significantly more likely to be reluctant to receive the vaccine.” However, nowhere in the manuscript were these two variables, people living in the city and having a family history, compared together. On the contrary, the authors showed that people with a family history of COVID-19 infection “were more likely to accept vaccination (line 157).

Comments on the Quality of English Language

The grammar must be revised.

Author Response

Thank you very much for taking the time to review this manuscript. Please find the detailed responses in the attached document. The revisions/corrections highlighted in the re-submitted manuscript 

Reviewer 2 Report

Comments and Suggestions for Authors

Dear Authors,

I think this is an interesting report that investigates the hesitancy and associated factors of COVID-19 vaccination in a rural Saudi Arabian community. However, I would like you to reiterate the following points.

Major Point

There is a statement opposite to the abstract and results.

The author states that those with a family history of COVID-19 were more likely to accept vaccination, but the abstract states that those with a family history of COVID-19 were significantly more likely to be hesitant to vaccinate. Please make the correct statement.

Material and Methods

An explanation of the social situation in Saudi Arabia at the time of the survey would be helpful to better understand the findings.

Please describe the prevalence of COVID-19 in Saudi Arabia at the time of the survey.

How many doses of vaccine were common in Saudi Arabia at the time of the survey?

Are the respondents to the online survey specific to the Jazan Province?

Results

The most common reason for hesitating to vaccinate was fear of vaccine side effects. How well informed are most people in Saudi Arabia about side effects? I think it makes a difference whether people are afraid because they don't have accurate information or because they have some information but are still afraid.

Discussion

While the vaccine acceptance rate was as high as 83.6% in the study of college students, the vaccine acceptance rate was lower among the college-educated respondents in this study. The author states that "further research is needed," but it would be desirable to have the author's own discussion on this issue.

Author Response

Thank you very much for taking the time to review this manuscript. Please find the detailed responses in the attached document. The revisions/corrections are highlighted in the re-submitted manuscript 

Reviewer 3 Report

Comments and Suggestions for Authors

Abstract:  there are some issues to address:
The abstract should be more concise. It includes unnecessary details about the WHO's COVID-19 declaration and the specific vaccines used in Saudi Arabia. Focus on the study's objectives, methods, major findings, and implications.
The sample size mentioned in the abstract (560 participants) differs from the sample size mentioned in the results section (569 participants). This inconsistency should be clarified and corrected.

Introduction is quite lengthy and could be more concise. The introduction should also provide a clear statement of the research question or hypothesis to guide the reader.
It would benefit from a more concise presentation of this information. The transition from the global context to Saudi Arabia's specific situation could be smoother.
the introduction mentions that the pandemic ended in May 2023, which might not be accurate or relevant, given the evolving nature of the COVID-19 situation. It's crucial to use up-to-date information, and the study's focus should remain on the vaccine hesitancy issue.
Methodology:
It lacks information on the sample size calculation and sampling method. The manuscript should clarify how the study participants were recruited from social media platforms and address potential bias from this recruitment method. Additionally, the use of Google Forms for data collection could lead to self-selection bias, and this limitation should be acknowledged.

Results:
There are some inconsistencies in the reported sample size (569 participants) compared to the earlier mentioned number (560 participants). Additionally, it's unclear why certain percentages are reported with two decimal places (e.g., 63.1%) while others have one decimal place (e.g., 36.9%).

Discussion could be structured more clearly. It would be beneficial to separate the discussion of vaccine acceptance rates from the discussion of factors influencing hesitancy. Additionally, discussing the implications of the study's findings for public health policy and vaccination campaigns in Saudi Arabia would be valuable.
the discussion could be more concise. It repeatedly mentions the acceptance and hesitancy percentages, which have already been presented in the results section. It would be more valuable to focus on the factors influencing vaccine hesitancy specific to the study population and implications for public health interventions. Additionally, the limitations of the study, such as the lack of a probability sampling method and the potential for selection bias, should be discussed more thoroughly.

Conclusion doesn't offer specific recommendations for addressing vaccine hesitancy in the Jazan region or Saudi Arabia, which would be beneficial for policymakers and readers. The need for further research is mentioned, but it could be more specific about what areas of research are required.

Comments on the Quality of English Language

There are several typos and grammatical mistakes in the manuscript, few examples are given below: 

In Abstract:
"vaccine hesitation causes" should be "reasons for vaccine hesitancy."
"in the Jazan Provence" should be "in the Jazan Province." Spelling mistake
"interventional strategies" should be "intervention strategies."

Line 235, "tele Visions" should be "televisions."

In Results:
"The Table 1 data shows" should be "The data in Table 1 show."
"acceptance of vaccination against COVID-19 infection" could be clearer as "acceptance of COVID-19 vaccination."

  1. Discussion:
    "A number of studies from Saudi Arabia have reported" could be rephrased as "Several studies from Saudi Arabia have reported."
    "lesser acceptance of COVID-19 vaccine among" should be "lower acceptance of the COVID-19 vaccine among."

    Conclusions:
    "about one-third of the study participants" could be clearer as "approximately one-third of the study participants."
    At the end of conclusion, why 6 is written after Saudi Arabia???

These are some of the English and grammatical mistakes in the manuscript. Careful proofreading and editing can help improve the clarity and professionalism of the text.

Author Response

(The authors gave the same response as above.)

Round 2

Reviewer 1 Report

Comments and Suggestions for Authors

-          The authors discuss results not presented in table 1, yet they refer to table 1 in line 141.

-          Table 2: the words are repeated (line 157) Table 2. Post Post-hoc analysis

-          There are multiple grammar and spelling mistakes in the text.

-          Text lines 160-168 are not comprehensible. Please simplify

-          Line 173 :” Table 2. Post-hoc analysis for the significant Chi-square results in Table 4 by using Bonferroni method.” what does this mean??

-          Tables 2 and 4 should be simplified (e.g. remove the total n, the adjusted residual and the repeated n%, ect..

-          In lines 159-167, the authors conducted a post hoc analysis and appropriately modified the results using the Bonferroni-corrected p value. Consequently, the obtained results were not statistically significant. So the researchers returned to their first analysis, which had been omitted from the text which was conducted using a 2X2 statistical framework. Kindly note that the 2X2 analysis does not provide sufficient evidence to support the results drawn. Nonetheless, the researchers incorporated these findings into their publication. The current situation is deemed to be unsatisfactory and does not meet the required standards.

Comments on the Quality of English Language

 The work still contains significant errors in grammar and spelling. The content has become incoherent and filled with conflicting information.

Author Response

Author's responses to reviewer’s comments and suggestions

Comments and Suggestions for Authors

-          The authors discuss results not presented in table 1, yet they refer to table 1 in line 141.

Thanks for the suggestion. This error is corrected

-          Table 2: the words are repeated (line 157) Table 2. Post Post-hoc analysis

Corrected

-          There are multiple grammar and spelling mistakes in the text.

Thank you for pointing this out. We have corrected again with additional help. The revised manuscript has several changes to correct the language errors.  

-          Text lines 160-168 are not comprehensible. Please simplify

We hope the revised paragraph provide more clarity. Three tables are added as supplementary tables for the readers to verify the multiple two-by-two analysis

-          Line 173 :” Table 2. Post-hoc analysis for the significant Chi-square results in Table 4 by using Bonferroni method.” what does this mean??

Corrected

-          Tables 2 and 4 should be simplified (e.g. remove the total n, the adjusted residual and the repeated n%, ect..

Done. Corrected

-          In lines 159-167, the authors conducted a post hoc analysis and appropriately modified the results using the Bonferroni-corrected p value. Consequently, the obtained results were not statistically significant. So the researchers returned to their first analysis, which had been omitted from the text which was conducted using a 2X2 statistical framework. Kindly note that the 2X2 analysis does not provide sufficient evidence to support the results drawn. Nonetheless, the researchers incorporated these findings into their publication. The current situation is deemed to be unsatisfactory and does not meet the required standards.

This is clarified in the paragraph starting line 162. 

Comments on the Quality of English Language

 The work still contains significant errors in grammar and spelling. The content has become incoherent and filled with conflicting information.

We have corrected the language issues with additional help. We are happy to take specific suggestions, if you have any

Reviewer 2 Report

Comments and Suggestions for Authors

The points we pointed out were considered and appropriately corrected.

Author Response

Thank you for the review. 

Reviewer 3 Report

Comments and Suggestions for Authors

The manuscript has been significantly improved.

Author Response

Thank you for the review and comments. We have improved the manuscript in the latest version